# Cardiovascular disease risk profile and management practices in 45 low-income and middle-income countries: A cross-sectional study of nationally representative individual-level survey data

David Peiris[1]*, Arpita Ghosh[2,3], Jennifer Manne-Goehler[4], Lindsay M. Jaacks[5], Michaela Theilmann[4], Maja E. Marcus[6], Zhaxybay Zhumadilov[7], Lindiwe Tsabedze[8], Adil Supiyev[9], Bahendeka K. Silver[10], Abla M. Sibai[11], Bolormaa Norov[12], Mary T. Mayige[13], Joao S. Martins[14], Nuno Lunet[15], Demetre Labadarios[16], Jutta M. A. Jorgensen[17], Corine Houehanou[18], David Guwatudde[19], Mongal S. Gurung[20], Albertino Damasceno[21], Krishna K. Aryal[22], Glennis Andall-Brereton[23], Kokou Agoudavi[24], Briar McKenzie[1], Jacqui Webster[1], Rifat Atun[5], Till Bärnighausen[4], Sebastian Vollmer[6], Justine I. Davies[25,26,27‡], Pascal Geldsetzer[4,28‡]

1 The George Institute for Global Health, UNSW Sydney, Sydney, New South Wales, Australia, 2 The George Institute for Global Health, UNSW Sydney, New Delhi, India, 3 Manipal Academy of Higher Education, Manipal, India, 4 Heidelberg Institute of Global Health, Medical Faculty and University Hospital, University of Heidelberg, Heidelberg, Germany, 5 Department of Global Health and Population, Harvard T.H. Chan School of Public Health, Boston, Massachusetts, United States of America, 6 Department of Economics and Centre for Modern Indian Studies, University of Goettingen, Goettingen, Germany, 7 Nazarbayev University School of Medicine, Nur-Sultan, Kazakhstan, 8 Eswatini Ministry of Health, Mbabane, Eswatini, 9 Laboratory of Epidemiology and Public Health, Center for Life Sciences, National Laboratory Astana, Nazarbayev University, Astana, Kazakhstan, 10 St. Francis Hospital Nsambya, Kampala, Uganda, 11 Department of Epidemiology & Population Health, Faculty of Health Sciences, American University of Beirut, Beirut, Lebanon, 12 National Center for Public Health, Ulaanbaatar, Mongolia, 13 National Institute for Medical Research, Dar es Salaam, Tanzania, 14 Faculty of Medicine and Health Sciences, Universidade Nacional Timor Lorosa'e, Dili, Timor-Leste, 15 Departamento de Ciências da Saúde Pública e Forenses e Educação Médica, Faculdade de Medicina da Universidade do Porto, Porto, Portugal, 16 Faculty of Medicine and Health Sciences, Stellenbosch University, Stellenbosch, South Africa, 17 Department of Public Health, University of Copenhagen, Copenhagen, Denmark, 18 Laboratory of Epidemiology of Chronic and Neurological Diseases, Faculty of Health Sciences, University of Abomey-Calavi, Cotonou, Benin, 19 Department of Epidemiology and Biostatistics, School of Public Health, Makerere University, Kampala, Uganda, 20 Health Research and Epidemiology Unit, Ministry of Health, Thimphu, Bhutan, 21 Faculty of Medicine, Eduardo Mondlane University, Maputo, Mozambique, 22 Monitoring Evaluation and Operational Research Project, Abt Associates, Kathmandu, Nepal, 23 Non-Communicable Diseases, Caribbean Public Health Agency, Port of Spain, Trinidad and Tobago, 24 Togo Ministry of Health, Lomé, Togo, 25 Institute of Applied Health Research, University of Birmingham, Birmingham, United Kingdom, 26 Centre for Global Surgery, Department of Global Health, Stellenbosch University, Cape Town, South Africa, 27 Medical Research Council/Wits University Rural Public Health and Health Transitions Research Unit, Faculty of Health Sciences, School of Public Health, University of the Witwatersrand, Johannesburg, South Africa, 28 Division of Primary Care and Population Health, Department of Medicine, Stanford University, Stanford, California, United States of America

‡ These authors are joint senior authors on this work.
* dpeiris@georgeinstitute.org

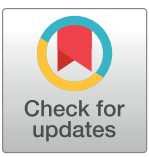

Data Availability Statement: Study data cannot be shared publicly because the data use agreements

## Abstract

### Background

Global cardiovascular disease (CVD) burden is high and rising, especially in low-income and middle-income countries (LMICs). Focussing on 45 LMICs, we aimed to determine (1)

prohibit this. The data analysis code is available here: https://doi.org/10.7910/DVN/RRMO8B. We have also included in the supporting information, additional details on accessing country-specific datasets (S3 Text).

**Funding:** This study was supported with funding from the Harvard McLennan Family Fund. DP is supported by fellowships from the National Health and Medical Research Council of Australia (1143904) and the Heart Foundation of Australia (101890). PG was supported by the National Center for Advancing Translational Sciences of the National Institutes of Health under Award Number KL2TR003143. The funding sources for this study had no involvement in the design, collection, analysis and interpretation of the data. The academic investigators participated in the design and oversight of the project. They had full access to all the data and had final responsibility for the decision to submit for publication. All authors gave approval to submit for publication.

**Competing interests:** The authors have declared that no competing interests exist.

**Abbreviations:** BMI, body mass index; BP, blood pressure; CVD, cardiovascular disease; DBP, diastolic blood pressure; GBD, Global Burden of Disease; IQR, interquartile range; LMICs, low-income and middle-income countries; NCD, non-communicable disease; RR, risk ratio; SBP, systolic blood pressure; STEPS, Stepwise Approach to Surveillance; WHO, World Health Organization; WHO PEN, World Health Organization Package of Essential Noncommunicable Disease Interventions.

the adult population's median 10-year predicted CVD risk, including its variation within countries by socio-demographic characteristics, and (2) the prevalence of self-reported blood pressure (BP) medication use among those with and without an indication for such medication as per World Health Organization (WHO) guidelines.

## Methods and findings

We conducted a cross-sectional analysis of nationally representative household surveys from 45 LMICs carried out between 2005 and 2017, with 32 surveys being WHO Stepwise Approach to Surveillance (STEPS) surveys. Country-specific median 10-year CVD risk was calculated using the 2019 WHO CVD Risk Chart Working Group non-laboratory-based equations. BP medication indications were based on the WHO Package of Essential Non-communicable Disease Interventions guidelines. Regression models examined associations between CVD risk, BP medication use, and socio-demographic characteristics. Our complete case analysis included 600,484 adults from 45 countries. Median 10-year CVD risk (interquartile range [IQR]) for males and females was 2.7% (2.3%–4.2%) and 1.6% (1.3%–2.1%), respectively, with estimates indicating the lowest risk in sub-Saharan Africa and highest in Europe and the Eastern Mediterranean. Higher educational attainment and current employment were associated with lower CVD risk in most countries. Of those indicated for BP medication, the median (IQR) percentage taking medication was 24.2% (15.4%–37.2%) for males and 41.6% (23.9%–53.8%) for females. Conversely, a median (IQR) 47.1% (36.1%–58.6%) of all people taking a BP medication were not indicated for such based on CVD risk status. There was no association between BP medication use and socio-demographic characteristics in most of the 45 study countries. Study limitations include variation in country survey methods, most notably the sample age range and year of data collection, insufficient data to use the laboratory-based CVD risk equations, and an inability to determine past history of a CVD diagnosis.

## Conclusions

This study found underuse of guideline-indicated BP medication in people with elevated CVD risk and overuse by people with lower CVD risk. Country-specific targeted policies are needed to help improve the identification and management of those at highest CVD risk.

### Author summary

#### Why was this study done?

- CVD burden in low-income and middle-income countries (LMICs) is high and rising.

- CVD risk estimation using validated risk prediction equations is recommended in most guidelines; however, there are few population-representative analyses of CVD risk and its association with socio-demographic characteristics.

- Despite guidelines recommending using CVD risk estimates as an essential first step in guiding management practices, the extent to which risk-based approaches are being implemented in LMICs is not well characterised.

## What did the researchers do and find?

- We analysed population-representative survey data from 45 LMICs to determine country-specific levels of CVD risk, associations between socio-demographic factors and levels of CVD risk, and adherence to WHO guidelines on use of blood pressure medication.

- We found high variation in CVD risk profiles, with higher levels of risk in the Europe and the Eastern Mediterranean region and lower levels of risk in sub-Saharan Africa, as well as an inverse association between CVD risk and higher education and employment in most countries.

- We found an underuse of medicines in people at elevated CVD risk across all countries (only 24.2% of males and 41.6% of females at high CVD risk are taking guideline-recommended BP medication) and an overuse of medicines in people at lower levels of CVD risk, with 47% of all BP medication being used by people at low CVD risk without a guideline indication for use.

## What do these findings mean?

- There is large variation in CVD risk across LMICs, and an inverse association between CVD risk and higher education and employment in most countries.

- There is an overuse of medicines in people at lower levels of CVD risk and an underuse of medicines in people at elevated CVD risk across all countries.

- The large heterogeneity of the findings in this study reflects varying country contexts. Country-specific targeted policies are needed to improve the identification and management of those at highest CVD risk.

## Introduction

Although cardiovascular disease (CVD) disease burden is declining in high-income countries, it is rising in low-income and middle-income countries (LMICs) and is the leading cause of death worldwide, accounting for an estimated 17.8 million deaths and an age-standardised death rate of 233 per 100,000 in 2017 [1]. CVD occurs at younger ages in LMICs than in high-income countries and exhibits strong socio-economic gradients, both in terms of disease burden and economic hardship in managing it [2]. Coordinated action with respect to CVD is especially important for several reasons: (1) the rising health and economic burden of CVD is placing considerable strain on the individuals affected by CVD, their families, and health systems more broadly; (2) the leading risk factors for CVD can be diagnosed and treated at relatively low cost compared with the cost of treating CVD events such as hospitalisation for myocardial infarction and stroke; and (3) better information on how CVD risk varies globally could inform health system planning and targeting of global and national CVD programmes and help accelerate progress in achieving Sustainable Development Goal 3.4 to reduce premature mortality from non-communicable diseases (NCDs) by one-third by 2030 [3–5].

The 2010 World Health Organization Package of Essential Noncommunicable Disease Interventions (WHO PEN) for primary healthcare in LMICs identifies a set of interventions

for strengthening primary healthcare to tackle NCDs through the use of low-cost medicines, tools, and technologies [6]. WHO PEN is the most commonly used technical guidance for NCDs in LMICs and has been implemented with varying degrees of success in several countries [7–10]. Other more recent technical guidance includes the 2016 Global Hearts Initiative, launched by WHO and the US Centers for Disease Control and Prevention, which comprises 5 technical packages of evidence-based interventions for the prevention and management of CVDs in primary healthcare [11]. Despite the availability of guidance, however, in LMICs implementation of routine national surveillance and cost-effective interventions to address CVD and its risk factors remains challenging [12,13].

About 60% of all cardiovascular deaths will occur in asymptomatic people who have not had a previous cardiovascular event [14]. The challenges of identifying people at high risk are extensive, especially in settings with low-skilled health workforces and inadequate access to healthcare services. Over the past decade, there has been a shift in CVD prevention from assessing single risk factor abnormalities to management based on a person's future risk of experiencing a cardiovascular event. This approach has been demonstrated to be superior to assessment of single risk factors when identifying who will benefit the most from treatment and is endorsed in WHO PEN and the Global Hearts Initiative [6,11,14].

In this paper, we analyse individual-level data from 45 nationally representative population-based surveys. Taking primarily a country-level health system perspective, we aim to (1) estimate CVD risk profiles, (2) determine management patterns in each country based on blood pressure (BP) medicine use as recommended by WHO PEN guidelines, and (3) determine whether wealth and educational status are associated with variation in CVD risk and management practices.

## Methods

This study is reported as per the Strengthening the Reporting of Observational Studies in Epidemiology (STROBE) guideline (S1 STROBE Checklist). A study outline, including proposed figures and tables, was developed prior to conducting the analyses. No changes were made to these analyses; however, an initial pre-planned analysis on BP control rates was excluded due to low numbers with available data.

### Data source

We retrieved datasets from nationally representative population-based surveys in 45 LMICs. The approach to identifying and obtaining these datasets has been described previously [15]. In brief, data were obtained from the Stepwise Approach to Surveillance (STEPS) surveys, and national surveys after 2005 based on a systematic review of the literature. The requirements for dataset inclusion in this study were as follows: (1) the survey was conducted in an upper-middle-, lower-middle-, or low-income country according to the World Bank country income groupings at the time the survey was conducted; (2) the survey was nationally representative with a response rate of 50% or greater; (3) the survey included individual-level data for people aged over 30 years; (4) the survey included availability of all the essential variables needed to estimate CVD risk (age, sex, smoking status, systolic BP [SBP] and BP medication use, body weight, and height, with a missing rate of no greater than 35% on any 1 of these variables); and (5) the survey included measures to ascertain the wealth and educational status of the participants. We included 32 STEPS surveys and 14 non-STEPS surveys in the analysis (46 surveys in 45 countries). The Zanzibar STEPS survey was analysed separately to the rest of Tanzania as it is a semi-autonomous region with a separate survey and ministry of health. The details of

countries included in the study and the construction of household wealth quintiles are presented in S1 Table and S1 Text, respectively.

## Estimation of CVD risk

For estimation of risk, we used the recently updated non-laboratory-based WHO CVD risk prediction equations, which is recommended in WHO PEN and the Global Hearts Initiative guidelines and calculates sex-specific 10-year risk of a CVD event defined as myocardial infarction or stroke [16]. These equations, derived by the WHO CVD Risk Chart Working Group, use age, smoking status, SBP, and body mass index (BMI), and are recalibrated to 21 regions using CVD incidence data from the Global Burden of Disease (GBD) studies. In almost all survey samples, data on cholesterol levels were not available, and in many country survey samples diabetes status was also not available. Consequently, the non-laboratory-based WHO risk equations were the more appropriate equations to use, given they do not rely on either of these variables. Although the non-laboratory-based equations discriminate similarly to the laboratory-based equations overall, they are known to underestimate risk in people with diabetes [16]. The equations were validated for people aged 40–80 years, but risk assessment is recommended by WHO PEN for younger people with risk factors. Consequently, we calculated risk for people aged 30–39 years assuming their age was 40 years. For people currently taking a BP medication, pre-treatment SBP and diastolic BP (DBP) levels were estimated using the following formulae described by Wald and Law [17,18]:

$$\mathrm{SBP_{pre-treatment}} - \mathrm{SBP_{treated}} = 9.1 + 0.10(\mathrm{SBP_{pre-treatment}} - 154)$$

$$\mathrm{DBP_{pre-treatment}} - \mathrm{DBP_{treated}} = 5.5 + 0.11(\mathrm{DBP_{pre-treatment}} - 97)$$

Because few country surveys had a specific question on whether participants had pre-existing CVD, we assumed the entire sample was CVD-free, which will lead to an underestimate of the population's CVD risk.

## Estimation of management gaps

The primary measure for assessing management gaps was BP medication use. Although management of all risk factors is recommended for people at elevated CVD risk, there is limited information collected on management of other CVD risk factors such as cholesterol and diabetes. Elevated CVD risk and indications for use of BP medication were based on WHO PEN guidelines, and indication for BP medication was defined as the presence of any of the following: (1) an extreme pre-treatment BP elevation (SBP $\geq$ 160 mm Hg or DBP $\geq$ 100 mm Hg), (2) a 10-year CVD risk $\geq$ 30%, or (3) a 10-year CVD risk of 20%–29% and a pre-treatment SBP $\geq$ 140 mm Hg or a pre-treatment DBP $\geq$ 90 mm Hg.

## Statistical analysis

The analysis plan is available in S2 Text. We estimated the prevalence of the main outcomes (CVD risk and BP medication use) using sampling weights that account for the complex sampling procedure. In addition, standard errors were adjusted for the stratified cluster sampling design. Each country sample was age-standardised to the 2017 world population profile as per 2017 GBD estimates [19]. For most surveys the upper age limit ranged from 64–74 years with the exception of 2 countries: India (49 years for females and 54 years for males) and Ecuador (59 years for both males and females). The median 10-year risk for each country was calculated based on the risk estimates for individuals with complete data in the age range 30–64 years

with the exception of India and Ecuador. Using the WHO CVD Risk Chart Working Group categories, we analysed the proportion of people in the following 10-year risk categories: <5%, 5% to <10%, 10% to <20%, 20% to <30%, and ≥30% [16]. When comparing variation between countries and when pooling data across countries, each country was weighted equally because the primary analysis unit of interest was a country's health system (regardless of the size of the population that it serves). Despite accounting for a large proportion of the total sample size, this approach meant that data from India did not influence the cross-country summary statistics and regression estimates more than the data from any of the other countries. For cross-country analyses, we also conducted a sensitivity analysis for which each country was weighted according to its population size in 2015 [20].

The association between the main outcomes and socio-demographic variables was estimated through regressions that used sampling weights and accounted for clustering at the level of the primary sampling unit. For the continuous outcome—10-year CVD risk—a linear mixed effects model (using a random intercept for each country) with logarithm of the risk as the dependent variable and the relative change was reported. For assessing socio-demographic associations with 10-year CVD risk, a multiple variable model was fitted with sex, marital status (married or cohabiting versus never married/separated/divorced/widowed), educational level (primary schooling or higher education versus no schooling), employment status (working in the previous 12 months versus not working in the previous 12 months), and household wealth quintile (upper 3 quintiles versus bottom 2 quintiles) as independent binary variables for countries that had a minimum of 5 observations in each category. For the binary outcome of BP treatment, a modified Poisson regression model was fitted, and the risk ratios (RRs) for receiving treatment by education, marital status, employment, and wealth status were analysed separately by sex and for those indicated and not indicated for BP medication.

## Ethics

Local survey teams obtained approval from local ethics committees and informed consent from participants prior to conducting the surveys. The consent processes for participation in the surveys are available in the programme manuals (S3 Text). This study was designated "not human subjects research" and was thus deemed not to require additional ethical approval by the institutional review board of the Harvard T.H. Chan School of Public Health on May 9, 2018.

## Results

### Sample characteristics

Complete case analysis was conducted for 600,484 people from 45 LMICs (Table 1). At least 1 of the variables needed to ascertain CVD risk was missing for 4.1% (n = 25,752) of the people in the overall sample, with this proportion varying by country (S1 Fig). These individuals were excluded from the analysis. Aside from female participants having a higher probability of being excluded due to missing data than male participants, the differences in participant characteristics between those included in the analysis and those excluded due to missing data were small (S2 Table).

The median (IQR) age for the total sample was 44.7 years (43.0–46.9), median (IQR) female proportion was 51.7% (49.5%–55.3%), median (IQR) proportion of current smokers was 18.9% (11.5%–25.3%), and median (IQR) BMI was 25.7 kg/m$^2$ (23.3–26.8). The proportion of individuals taking BP medication varied widely across countries—from 0.6% in Vanuatu to 25.5% in Belarus (median = 8.6%; IQR = 4.1%–14.2%). The median (IQR) SBP among people not taking and taking BP medication was 124.3 mm Hg (123.0–127.4) and 144.8 mm Hg

**Table 1. Weighted distribution of cardiovascular disease risk factors in participants aged 30–74 years from population-based surveys conducted in 45 low- and middle-income countries between 2005 and 2017.**

| Country | Sex | Sample size[¶] | Median age (years) | Age range (years) | Current smoker (%) | Taking BP medication (%) | Median SBP for participants not taking BP medication (IQR) | Median SBP for participants taking BP medication (IQR) | Median BMI (IQR) |
|---|---|---|---|---|---|---|---|---|---|
| **Latin America and the Caribbean** | | | | | | | | | |
| Belize | Female | 818 | 41 | 30–74 | 2.7 | 8.8 | 110.0 (100.0, 123.0) | 133.7 (115.5, 156.0) | 29.5 (25.7, 33.8) |
| | Male | 636 | 43 | 30–74 | 25.4 | 4.9 | 119.4 (109.5, 133.0) | 140.0 (120.4, 153.3) | 26.8 (23.3, 30.4) |
| Brazil | Female | 23,134 | 47 | 30–74 | 13.1 | 25.1 | 120.0 (110.0, 130.0) | 134.5 (123.0, 148.0) | 26.8 (23.7, 30.7) |
| | Male | 17,682 | 46 | 30–74 | 20.2 | 17.0 | 126.0 (118.0, 136.0) | 135.5 (125.5, 149.0) | 26.2 (23.6, 29.1) |
| Chile | Female | 1,994 | 46 | 30–74 | 32.8 | 15.4 | 119.0 (109.0, 132.0) | 137.5 (124.0, 152.4) | 28.1 (25.1, 32.0) |
| | Male | 1,350 | 46 | 30–74 | 38.2 | 7.3 | 127.0 (118.3, 139.5) | 146.0 (135.6, 159.3) | 27.3 (24.9, 29.7) |
| Costa Rica | Female | 1,779 | 44 | 30–74 | 6.1 | 28.7 | 110.0 (100.0, 120.0) | 120.0 (110.0, 133.5) | 27.6 (24.1, 31.4) |
| | Male | 661 | 44 | 30–74 | 22.6 | 17.2 | 115.0 (100.0, 120.0) | 120.0 (119.1, 139.3) | 26.2 (23.4, 29.0) |
| Ecuador | Female | 11,345 | 42 | 30–59 | 7.7 | 10.2 | 116.0 (108.0, 123.5) | 130.5 (120.0, 145.3) | 27.9 (25.0, 31.3) |
| | Male | 8,107 | 42 | 30–59 | 39.9 | 5.1 | 121.0 (114.5, 129.0) | 133.6 (123.5, 146.0) | 26.7 (24.3, 29.2) |
| Grenada | Female | 547 | 43 | 30–64 | 7.1 | 23.6 | 122.5 (113.5, 135.0) | 142.7 (129.1, 158.8) | 28.3 (24.8, 33.1) |
| | Male | 363 | 42 | 30–64 | 32.2 | 11.4 | 130.0 (120.5, 142.5) | 144.2 (135.0, 154.6) | 25.1 (22.2, 27.9) |
| Guyana | Female | 1,135 | 44 | 30–69 | 3.7 | 17.7 | 122.0 (113.5, 134.5) | 138.5 (121.2, 151.5) | 28.4 (24.4, 32.6) |
| | Male | 797 | 44 | 30–69 | 31.5 | 10.9 | 125.5 (118.0, 137.0) | 142.8 (131.5, 159.2) | 24.9 (21.9, 28.5) |
| Mexico | Female | 6,138 | 45 | 30–74 | 6.2 | 13.6 | 117.0 (106.5, 129.5) | 136.5 (121.5, 152.5) | 29.1 (25.7, 32.9) |
| | Male | 4,282 | 48 | 30–74 | 21.8 | 9.0 | 127.0 (117.0, 136.5) | 146.5 (129.5, 160.5) | 27.7 (25.0, 30.6) |
| StVG | Female | 1,439 | 43 | 30–69 | 2.6 | 18.6 | 121.5 (112.0, 133.5) | 140.8 (125.6, 157.5) | 29.6 (25.6, 34.7) |
| | Male | 1,228 | 43 | 30–70 | 24.2 | 7.2 | 126.0 (115.0, 137.5) | 140.9 (126.3, 157.1) | 24.8 (22.1, 28.4) |
| **Europe and the Eastern Mediterranean** | | | | | | | | | |
| Albania | Female | 2,073 | 40 | 30–49 | 3.4 | 4.2 | 129.0 (122.7, 135.7) | 146.9 (134.7, 164.9) | 25.5 (23.1, 28.5) |
| | Male | 1,600 | 41 | 30–49 | 52.6 | 1.7 | 133.7 (128.3, 140.7) | 144.4 (136.1, 152.7) | 26.4 (24.8, 28.5) |
| Azerbaijan | Female | 1,323 | 46 | 30–69 | 0.2 | 17.7 | 122.5 (112.4, 133.0) | 148.0 (133.3, 162.5) | 27.8 (24.4, 31.6) |
| | Male | 905 | 45 | 30–69 | 51.7 | 11.0 | 124.5 (117.0, 135.0) | 150.0 (136.6, 164.1) | 26.4 (24.2, 29.3) |
| Belarus | Female | 2,550 | 49 | 30–69 | 12.2 | 31.2 | 126.0 (118.0, 138.3) | 150.0 (139.0, 165.0) | 27.9 (24.1, 32.0) |
| | Male | 1,754 | 47 | 30–69 | 48.5 | 18.9 | 132.5 (124.0, 144.5) | 154.4 (142.2, 168.0) | 26.9 (24.3, 29.9) |

*(Continued)*

**Table 1.** (Continued)

| Country | Sex | Sample size⁵ | Median age (years) | Age range (years) | Current smoker (%) | Taking BP medication (%) | Median SBP for participants not taking BP medication (IQR) | Median SBP for participants taking BP medication (IQR) | Median BMI (IQR) |
|---|---|---|---|---|---|---|---|---|---|
| Georgia | Female | 2,425 | 50 | 30–70 | 7.1 | 23.9 | 122.5 (112.5, 135.7) | 144.0 (130.0, 162.5) | 29.0 (24.8, 33.8) |
| | Male | 979 | 48 | 30–70 | 55.3 | 16.4 | 126.5 (119.5, 140.0) | 150.0 (131.9, 168.0) | 28.0 (24.7, 31.5) |
| Kazakhstan | Female | 4,298 | 50 | 30–74 | 2.6 | 24.0 | 122.0 (116.5, 128.0) | 142.5 (133.5, 154.6) | 25.7 (22.8, 28.9) |
| | Male | 3,072 | 47 | 30–74 | 43.9 | 13.9 | 125.0 (120.0, 130.5) | 142.5 (135.0, 154.0) | 25.4 (23.2, 27.7) |
| Kyrgyzstan | Female | 1,379 | 45 | 30–64 | 3.1 | 15.0 | 127.0 (117.5, 140.0) | 157.5 (142.2, 174.8) | 27.7 (24.4, 31.6) |
| | Male | 809 | 44 | 30–64 | 47.8 | 7.3 | 130.5 (121.5, 143.5) | 161.0 (140.2, 181.4) | 25.7 (22.7, 29.4) |
| Lebanon | Female | 957 | 41 | 30–74 | 34.6 | 14.1 | 120.0 (110.0, 127.5) | 131.0 (120.9, 145.2) | 27.2 (23.7, 31.5) |
| | Male | 805 | 45 | 30–74 | 49.2 | 14.2 | 129.6 (120.0, 140.0) | 140.0 (126.2, 160.0) | 28.0 (25.0, 30.8) |
| Moldova | Female | 2,314 | 49 | 30–69 | 4.8 | 18.5 | 128.0 (119.5, 143.5) | 155.0 (142.9, 172.3) | 27.9 (24.2, 32.4) |
| | Male | 1,397 | 46 | 30–69 | 43.2 | 11.4 | 131.5 (123.5, 145.5) | 156.5 (139.2, 170.2) | 26.6 (23.8, 30.0) |
| Russian Federation | Female | 1,939 | 48 | 30–74 | 9.4 | 20.8 | 123.0 (112.5, 135.0) | 145.0 (134.5, 159.5) | 26.0 (22.6, 30.2) |
| | Male | 1,137 | 48 | 30–74 | 45.4 | 18.0 | 128.5 (120.0, 135.0) | 157.5 (141.3, 170.0) | 25.9 (24.2, 29.2) |
| Tajikistan | Female | 1,162 | 37 | 30–70 | 0.1 | 12.2 | 125.5 (116.5, 137.0) | 155.0 (140.0, 164.7) | 26.6 (23.2, 30.5) |
| | Male | 782 | 42 | 30–70 | 10.3 | 7.2 | 132.0 (123.5, 143.0) | 151.5 (137.5, 170.0) | 26.0 (23.4, 29.0) |
| **Southeast Asia and the western Pacific** | | | | | | | | | |
| Bhutan | Female | 1,251 | 41 | 30–69 | 3.2 | 8.8 | 123.0 (114.0, 136.0) | 144.5 (128.0, 162.6) | 24.4 (22.1, 27.4) |
| | Male | 867 | 39 | 30–69 | 8.0 | 4.5 | 125.0 (117.6, 135.5) | 147.2 (126.7, 163.7) | 23.4 (21.6, 25.8) |
| Cambodia | Female | 2,926 | 43 | 30–64 | 6.9 | 5.3 | 110.5 (103.0, 120.5) | 129.5 (116.5, 144.1) | 21.8 (19.6, 24.5) |
| | Male | 1,578 | 42 | 30–64 | 57.5 | 2.4 | 118.0 (110.0, 127.5) | 130.8 (119.8, 145.4) | 21.3 (19.8, 23.3) |
| China | Female | 4,183 | 51 | 30–74 | 3.8 | 11.1 | 120.0 (110.0, 130.0) | 144.0 (131.5, 160.0) | 23.3 (21.2, 25.8) |
| | Male | 3,729 | 51 | 30–74 | 57.6 | 9.8 | 121.0 (114.0, 131.0) | 141.0 (131.0, 154.0) | 23.4 (21.2, 25.7) |
| India | Female | 320,763 | 38 | 30–49 | 10.7 | 4.2 | 116.7 (108.3, 126.0) | 130.0 (117.3, 144.3) | 22.6 (19.8, 25.8) |
| | Male | 57,737 | 40 | 30–54 | 56.2 | 3.3 | 122.3 (114.3, 130.7) | 135.3 (122.0, 149.0) | 22.6 (20.1, 25.2) |
| Indonesia | Female | 10,884 | 48 | 30–74 | 2.7 | 6.4 | 128.0 (116.5, 145.5) | 159.0 (140.5, 177.8) | 24.6 (21.6, 27.7) |
| | Male | 9,973 | 49 | 30–74 | 66.9 | 3.3 | 129.0 (119.5, 141.5) | 157.5 (138.5, 178.5) | 22.2 (19.9, 25.1) |
| Mongolia | Female | 1,537 | 41 | 30–64 | 55.7 | 9.5 | 130.0 (120.5, 142.0) | 152.2 (137.0, 163.4) | 25.2 (22.5, 28.5) |

(*Continued*)

**Table 1.** (Continued)

| Country | Sex | Sample size⁵ | Median age (years) | Age range (years) | Current smoker (%) | Taking BP medication (%) | Median SBP for participants not taking BP medication (IQR) | Median SBP for participants taking BP medication (IQR) | Median BMI (IQR) |
|---|---|---|---|---|---|---|---|---|---|
| | Male | 2,255 | 41 | 30–64 | 7.8 | 17.0 | 121.5 (113.0, 132.5) | 140.4 (127.0, 159.5) | 25.9 (23.0, 29.5) |
| Nepal | Female | 2,110 | 44 | 30–69 | 17.0 | 5.7 | 124.5 (115.0, 138.0) | 147.1 (136.5, 158.9) | 22.6 (20.2, 25.5) |
| | Male | 1,040 | 45 | 30–69 | 32.7 | 6.4 | 130.5 (120.7, 142.5) | 144.7 (136.6, 161.8) | 22.6 (20.4, 25.4) |
| Timor-Leste | Female | 1,024 | 44 | 30–69 | 8.7 | 6.2 | 123.0 (113.5, 135.5) | 145.8 (129.0, 163.4) | 21.0 (19.0, 23.3) |
| | Male | 826 | 46 | 30–69 | 65.0 | 5.2 | 124.0 (115.0, 136.5) | 136.5 (117.5, 161.9) | 20.5 (18.8, 22.5) |
| Vanuatu | Female | 1,815 | 41 | 30–64 | 3.5 | 0.8 | 127.0 (116.0, 140.5) | 149.6 (133.7, 165.6) | 26.5 (23.2, 30.2) |
| | Male | 1,867 | 42 | 30–64 | 42.4 | 0.4 | 130.5 (121.0, 142.5) | 155.0 (145.7, 155.9) | 25.0 (22.6, 28.2) |
| **Africa** | | | | | | | | | |
| Algeria | Female | 2,836 | 43 | 30–69 | 0.6 | 13.3 | 124.5 (115.0, 136.5) | 141.0 (126.5, 157.0) | 28.2 (24.8, 32.0) |
| | Male | 2,356 | 43 | 30–69 | 30.0 | 6.1 | 127.0 (118.0, 138.0) | 146.0 (132.4, 160.3) | 25.7 (23.0, 28.7) |
| Benin | Female | 1,709 | 40 | 30–69 | 0.6 | 4.1 | 121.5 (110.0, 132.5) | 163.6 (141.5, 180.3) | 23.0 (20.5, 26.0) |
| | Male | 1,645 | 42 | 30–69 | 14.8 | 3.1 | 126.0 (116.5, 140.0) | 155.4 (138.0, 174.2) | 22.5 (20.2, 24.8) |
| Botswana | Female | 1,672 | 43 | 30–69 | 5.9 | 18.4 | 123.5 (113.5, 138.0) | 139.5 (126.0, 151.6) | 26.2 (22.1, 30.8) |
| | Male | 742 | 41 | 30–69 | 38.6 | 9.9 | 128.0 (120.5, 141.0) | 143.0 (129.4, 162.1) | 22.4 (19.6, 25.7) |
| Burkina Faso | Female | 1,640 | 40 | 30–64 | 0.1 | 2.8 | 118.5 (109.5, 129.0) | 146.1 (122.8, 158.9) | 21.5 (19.3, 24.5) |
| | Male | 1,669 | 42 | 30–64 | 22.7 | 1.1 | 122.0 (113.5, 132.0) | 149.8 (131.0, 162.7) | 22.0 (20.1, 24.2) |
| Comoros | Female | 2,939 | 40 | 30–64 | 2.7 | 6.6 | 122.5 (111.5, 137.0) | 146.2 (128.4, 169.6) | 25.6 (22.2, 29.7) |
| | Male | 1,238 | 42 | 30–64 | 24.2 | 3.2 | 126.5 (116.5, 137.6) | 145.0 (127.6, 171.4) | 23.0 (20.9, 25.7) |
| Ghana | Female | 1,843 | 43 | 30–74 | 4.0 | 4.8 | 124.0 (113.5, 138.0) | 151.1 (141.0, 170.7) | 24.7 (21.1, 28.4) |
| | Male | 2,206 | 45 | 30–74 | 12.7 | 4.6 | 125.5 (114.0, 141.0) | 152.3 (134.0, 163.5) | 22.2 (20.0, 25.1) |
| Kenya | Female | 1,688 | 41 | 30–69 | 1.2 | 4.8 | 123.0 (113.0, 135.5) | 133.4 (128.0, 152.0) | 24.2 (20.8, 28.3) |
| | Male | 1,213 | 41 | 30–69 | 24.2 | 1.7 | 125.5 (117.0, 137.5) | 136.9 (131.3, 168.2) | 21.7 (19.5, 24.8) |
| Lesotho | Female | 1,270 | 37 | 30–49 | 0.3 | 15.5 | 120.0 (112.3, 130.0) | 138.0 (124.0, 160.4) | 26.9 (22.9, 31.3) |
| | Male | 1,185 | 40 | 30–59 | 49.7 | 7.2 | 122.7 (115.0, 132.0) | 132.8 (124.3, 146.6) | 21.7 (19.8, 24.6) |
| Liberia | Female | 685 | 38 | 30–64 | 3.4 | 6.0 | 122.5 (112.0, 137.0) | 150.0 (132.5, 172.2) | 27.0 (22.8, 32.9) |
| | Male | 583 | 39 | 30–64 | 18.7 | 4.6 | 125.5 (117.0, 138.0) | 144.5 (122.5, 179.2) | 24.4 (22.0, 28.0) |

(*Continued*)

**Table 1.** (Continued)

| Country | Sex | Sample size¶ | Median age (years) | Age range (years) | Current smoker (%) | Taking BP medication (%) | Median SBP for participants not taking BP medication (IQR) | Median SBP for participants taking BP medication (IQR) | Median BMI (IQR) |
|---|---|---|---|---|---|---|---|---|---|
| Morocco | Female | 2,606 | 46 | 30–74 | 0.4 | 10.5 | 127.0 (117.5, 139.0) | 146.0 (133.5, 163.7) | 28.1 (24.7, 32.0) |
| | Male | 1,423 | 45 | 30–74 | 25.4 | 4.2 | 127.5 (118.5, 138.0) | 148.6 (137.1, 165.2) | 25.0 (22.0, 28.0) |
| Mozambique | Female | 1,356 | 40 | 30–64 | 11.2 | 3.7 | 129.0 (116.5, 144.5) | 141.7 (129.7, 160.7) | 21.8 (19.8, 24.6) |
| | Male | 1,007 | 43 | 30–64 | 37.8 | 1.2 | 132.0 (120.0, 145.0) | 150.2 (142.6, 165.5) | 20.8 (19.0, 22.7) |
| Namibia | Female | 2,022 | 46 | 35–64 | 10.1 | 18.7 | 121.5 (110.0, 135.5) | 135.0 (122.0, 150.3) | 24.9 (21.1, 30.0) |
| | Male | 1,443 | 45 | 35–64 | 26.2 | 13.5 | 124.5 (115.0, 140.0) | 140.0 (128.5, 158.0) | 21.8 (19.4, 25.7) |
| Sudan | Female | 3,097 | 41 | 30–69 | 0.7 | 9.1 | 126.0 (116.0, 139.0) | 149.0 (133.5, 165.0) | 24.5 (20.8, 29.1) |
| | Male | 1,957 | 42 | 30–69 | 16.6 | 3.9 | 128.5 (119.5, 139.5) | 147.8 (130.7, 163.3) | 22.9 (19.9, 25.8) |
| Swaziland | Female | 1,220 | 42 | 30–70 | 1.7 | 14.8 | 123.0 (114.0, 138.0) | 146.0 (131.2, 169.4) | 29.3 (24.8, 33.8) |
| | Male | 590 | 42 | 30–69 | 18.5 | 6.1 | 124.5 (116.0, 136.0) | 138.9 (132.1, 155.6) | 24.1 (21.2, 27.7) |
| Tanzania | Female | 2,422 | 40 | 30–64 | 3.4 | 3.0 | 124.5 (115.0, 138.0) | 153.5 (126.6, 169.8) | 23.4 (20.5, 27.5) |
| | Male | 2,157 | 42 | 30–65 | 28.3 | 1.2 | 128.0 (119.0, 138.5) | 171.1 (138.3, 187.2) | 21.1 (19.2, 23.4) |
| Togo | Female | 1,183 | 40 | 30–64 | 3.4 | 4.5 | 121.5 (110.0, 137.0) | 140.5 (126.1, 157.7) | 23.5 (20.6, 27.7) |
| | Male | 1,219 | 40 | 30–64 | 19.2 | 1.3 | 126.5 (116.5, 137.5) | 139.8 (123.5, 180.3) | 22.1 (20.3, 24.1) |
| Uganda | Female | 1,324 | 42 | 30–69 | 5.3 | 4.4 | 122.5 (112.5, 136.0) | 152.5 (132.0, 167.2) | 22.9 (20.4, 26.3) |
| | Male | 920 | 41 | 30–69 | 23.2 | 1.3 | 124.5 (116.0, 136.5) | 135.8 (127.3, 153.0) | 21.4 (19.4, 23.4) |
| Zanzibar | Female | 1,193 | 40 | 30–64 | 0.9 | 2.8 | 126.5 (114.4, 145.0) | 153.5 (134.5, 197.3) | 24.6 (21.3, 29.7) |
| | Male | 766 | 40 | 30–64 | 15.0 | 1.7 | 131.0 (119.5, 146.4) | 152.7 (145.1, 175.8) | 22.9 (20.4, 26.1) |

All values are weighted except for sample size and age range. Forty-five countries were included, but there were 46 surveys in total: Zanzibar was surveyed separately from Tanzania.

¶Number of participants with non-missing predictors in World Health Organization cardiovascular disease risk charts—age, sex, current smoking status, body mass index, and SBP.

BP, blood pressure; IQR, interquartile range; SBP, systolic blood pressure; StVG, Saint Vincent and the Grenadines.

(139.8–149.9), respectively. For the latter group, the pre-treatment median (IQR) SBP was 153.8 mm Hg (148.3–159.6), using the Wald and Law adjustments.

## CVD risk distribution

The distribution of CVD risk, age-standardised using the GBD project's 2017 global population, among people aged 30 to 64 years for the 45 countries is shown in Fig 1. The median (IQR) 10-year CVD risk overall was 2.7% (2.3%–4.2%) for males and 1.6% (1.3%–2.1%) for

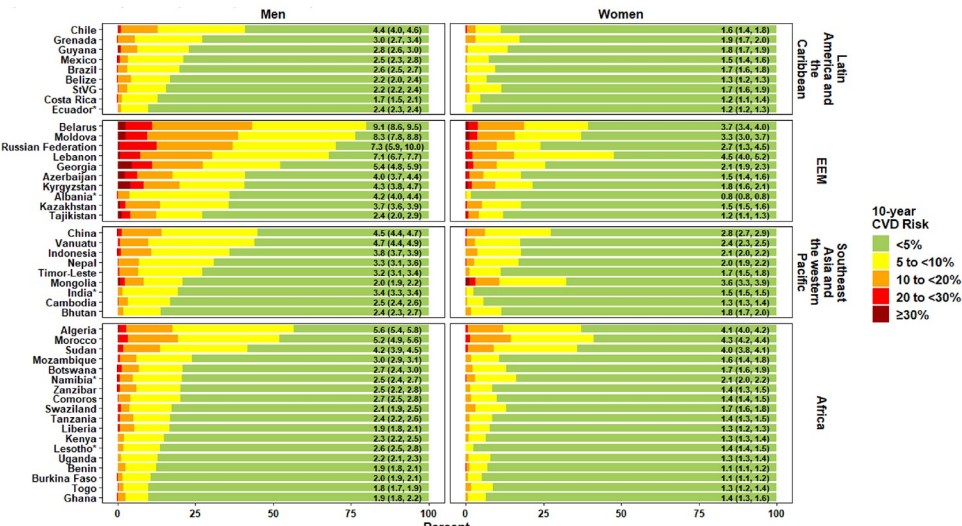

**Fig 1. CVD risk profile by country for men and women aged 30–64 years. The numbers in the green section of the stacked bars represent the median 10-year CVD risk along with the interquartile range for the country.** All estimates are age-standardised using the Global Burden of Disease project's 2017 global population. *The age range included in the sample varied in these 5 countries: India—30–49 years for women and 30–54 years for men; Ecuador—30–59 years for both men and women; Namibia—35–64 years for both men and women; Lesotho—30–49 years for women and 30–59 years for men; Albania—30–49 years for both men and women. CVD, cardiovascular disease; EEM, Europe and the Eastern Mediterranean; StVG, Saint Vincent and the Grenadines.

females, with wide variation across countries and regions. In a sensitivity analysis, risk was calculated excluding those under 40 years of age, and median (IQR) risk increased to 4.0% (3.4%–5.9%) for males and to 2.8% (2.2%–3.4%) for females (S2 Fig). Risk estimates tended to be lower in sub-Saharan Africa compared with countries in other world regions and were highest in Europe and the Eastern Mediterranean. The median proportion (IQR) of individuals overall at very low (<5%), low (5% to <10%), medium (10% to <20%), high (20% to <30%), and very high risk (≥30%) was 79.0% (59.1%–83.4%), 16.6% (12.4%–25.1%), 5.3% (2.6%–11.2%), 0.6% (0.2%–1.6%), and 0.1% (0.0%–0.6%), respectively, for males, and 88.5% (82.1%–92.5%), 9.7% (6.5%–14.2%), 1.8% (0.8%–4.6%), 0.1% (0.0%–0.6%), and 0.0% (0.0%–0.1%), respectively, for females.

Fig 2 examines the variation in 10-year CVD risk by education, wealth, marital status, and employment status from the multivariable regression. Having higher levels of education, working in the previous 12 months, and being married were consistently associated with lower CVD risk in most countries. There was a more mixed picture for household wealth, particularly with some sub-Saharan African countries showing higher median CVD risk for the wealthier quintiles. The magnitude of socio-demographic associations was broadly similar for females (S3 Fig) and males (S4 Fig), and there was a mixed contribution of individual risk factors driving these gradients (S3 Table) across all regions.

## Use of BP medication by CVD risk category

The median (IQR) percentage of people overall indicated for BP medication who were taking a BP medication was 24.2% (15.4%–37.2%) for males and 41.6% (23.9%–53.8%) for females. There was, however, a large variation in this pattern by country (Fig 3).

When compared with females at low CVD risk who were not indicated for BP medication, the univariable RR (95% CI) of receiving BP medication ranged from 2.1 (1.6–2.9) in Costa

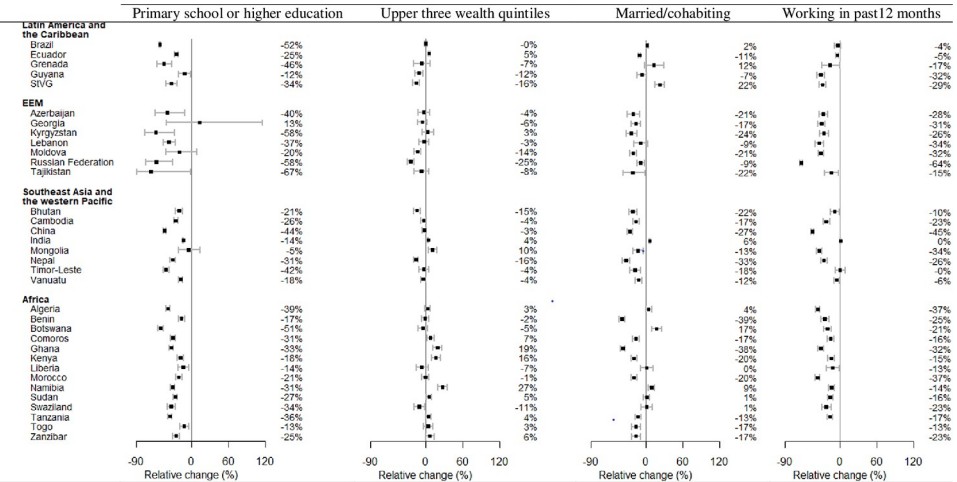

**Fig 2. Relative change (%) in 10-year CVD risk by educational level, household wealth, marital status, and employment status, from multivariable regression.** These are estimates from linear mixed models with the primary sampling units as the clusters. The outcome is logarithm of CVD risk, and the predictors are sex, educational level (primary school or higher education versus no schooling) household wealth quintile (middle/richer/richest versus poorer/poorest), marital status (married/cohabiting versus never married/separated/divorced/widowed), and employment status in the last 12 months (working versus not working). The estimates for Grenada and Morocco are based on linear regression as there are no primary sampling units and a single participant was sampled from each household. The countries with estimates not plotted either had a predictor missing or there were fewer than 5 participants in a category for 1 or more predictors. CVD, cardiovascular disease; EEM, Europe and the Eastern Mediterranean; StVG, Saint Vincent and the Grenadines.

Rica to 15.5 (9.8–24.6) in Albania among females. For males, the corresponding RR (95% CI) ranged from 2.8 (1.8–4.4) in Costa Rica to 16.5 (6.6–41.0) in Burkina Faso. Few countries had an association between BP medication use and socio-demographic characteristics regardless of whether BP medication use was indicated (S5 Fig) or not (S6 Fig).

Although a greater proportion of people at elevated CVD risk are taking BP medication compared with those at low CVD risk, almost half of survey participants who were taking BP medication overall were individuals at low CVD risk in most countries (Fig 4). Of all people taking BP medication, the median (IQR) proportion who were at low CVD risk and not indicated for BP medication was 47.1% (36.1%–58.6%), with minimal difference between males (45.1% [32.9%–57.3%]) and females (48.3% [37.2%–59.4%]).

## Discussion

In this study of 600,484 adults from 45 LMICs, we estimate the CVD risk profile using the recently published WHO region-specific, recalibrated risk prediction equations. Overall, we observed large variations in risk profile; an inverse association between CVD risk and higher education and employment in most countries; an overuse of medicines in people at lower levels of CVD risk; and an underuse of medicines in people at elevated CVD risk across all countries.

The large variation in risk profiles across and within countries is due to considerable variation in the presence of CVD risk factors. Countries with the highest median CVD risk tended to be middle-income countries. Females tended to have lower median CVD risk than males in almost all countries, which, in part, reflects the different coefficients used in sex-specific equations. These findings are consistent with those from the WHO CVD Risk Chart Working Group using the same equations for 79 countries (the majority of which are LMICs) [16] and

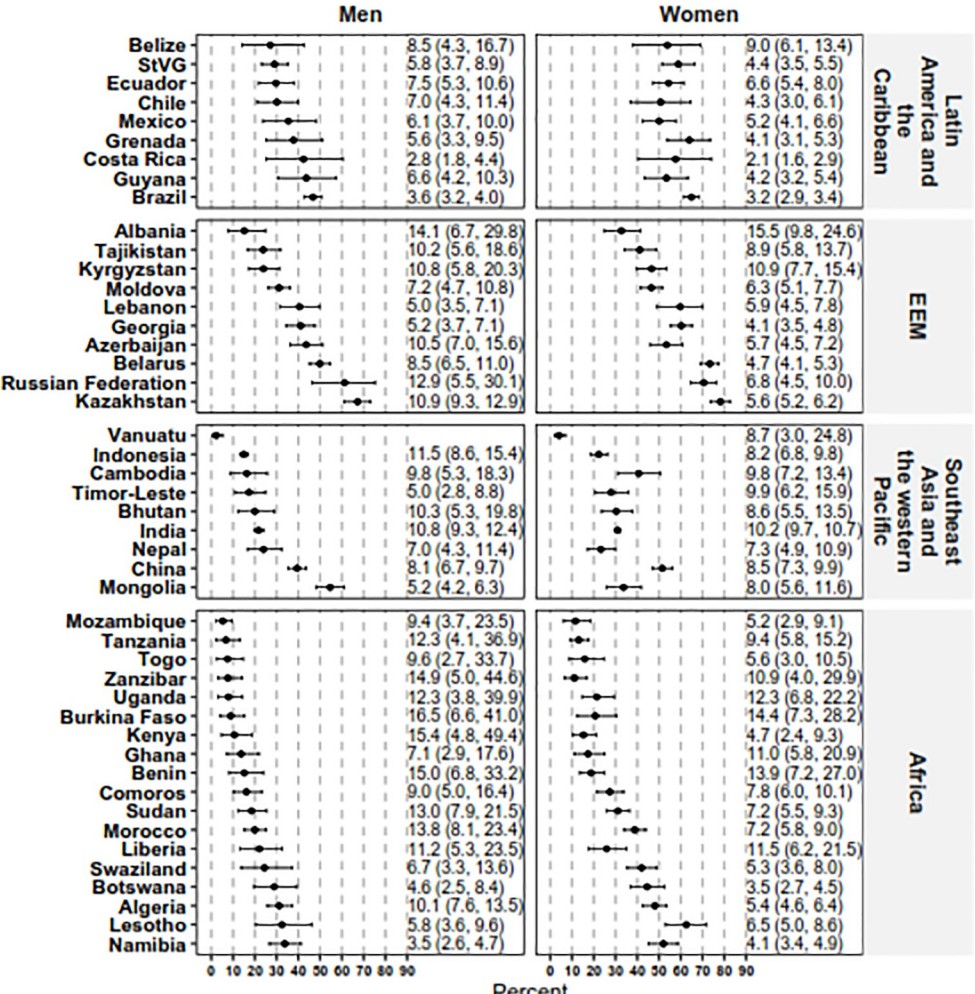

**Fig 3. Percentage of people indicated for BP medication who were taking medication, by sex.** The numbers on the right for each country and sex present univariate risk ratio of taking BP medication for an individual indicated for medication per World Health Organization/International Society of Hypertension guidelines, compared to an individual not indicated for medication. Indication for use of BP medication was defined as the presence of any of the following: an extreme blood pressure elevation (systolic BP $\geq$ 160 mm Hg or diastolic BP $\geq$ 100 mm Hg), a 10-year CVD risk $\geq$ 30%, or a 10-year CVD risk of 20%–29% and elevated blood pressure (systolic BP $\geq$ 140 mm Hg or diastolic BP $\geq$ 90 mm Hg). The risk ratio for men in Vanuatu was not calculated as among men indicated for medication, fewer than 5 men were on medication. BP, blood pressure; CVD, cardiovascular disease; EEM, Europe and the Eastern Mediterranean; StVG, Saint Vincent and the Grenadines.

those from Ueda et al. who used a different risk prediction model (Globorisk) in 10 high-, middle- and, low-income countries [21].

An important new finding from this study is the inverse socio-demographic gradients in CVD risk profiles, which appear to be driven by moderate elevations of multiple risk factors in all world regions. Although CVD risk levels tended to be higher overall in middle-income countries among the 45 countries studied, this study highlights the need to depart from narrowly conceptualising CVD as a disease of affluence and to apply an equity lens to implementing CVD risk programmes both within and across countries. Given that males tend to underutilise health services and people with lower education, lower household wealth, and unemployment may experience access barriers to high-quality care, intensified efforts to increase coverage and quality of care for these populations is warranted.

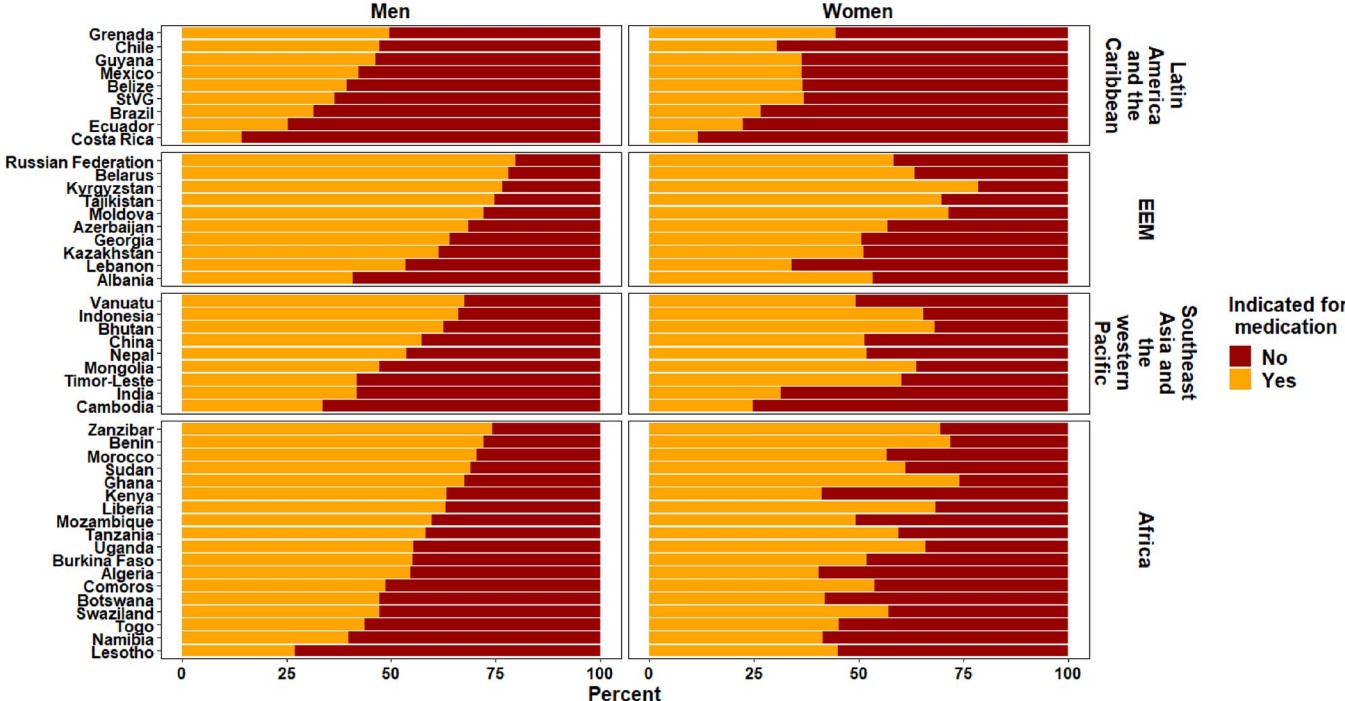

**Fig 4. Proportion of people taking blood pressure medication who were indicated/not indicated for medication based on guideline recommendations.**
EEM, Europe and the Eastern Mediterranean; StVG, Saint Vincent and the Grenadines.

The other key study finding was that around one-half of all BP medication (47.1%) is being used by people at lower levels of CVD risk, and only 24.2% of males and 41.6% of females at high risk were taking guideline-recommended medication. Given that a BP treatment strategy based on predicted CVD risk is more effective than one based on BP levels alone [14,22–24], the pattern of relative overuse in people at low CVD risk and undertreatment of people at high CVD risk (especially for men) represents inefficient use of scarce resources for many countries. An important explanation for this pattern is the likely persistence of single-risk-factor-based treatment over risk-based treatment. Many countries have local or regional hypertension guidelines, and these guidelines, many of which have been in operation for decades, generally recommend that if patients with hypertension do not respond to lifestyle interventions (diet, weight loss, exercise) within 3–6 months, they should be treated with an antihypertensive agent. Given the superiority of risk-based management, there needs to be harmonisation of conflicting guidelines and a shift toward targeting treatment to individuals who will gain the greatest benefit in terms of CVD events avoided. Factors influencing appropriate use of medicines are complex and go beyond improved risk stratification. Health systems barriers related to unreliable supply chains, healthcare providers (e.g., an insufficiently trained and supported workforce), and patients (e.g., financial barriers and other factors influencing non-adherence) all need to be considered [25,26].

There are a number of caveats to the study findings. First, the year of survey varied for the countries included in the study, and given rapid demographic transitions in many countries, the findings are most relevant to the year in which the country survey was conducted. Second, in some countries, most notably India, the upper age range limit of the sample was less than 65 years. Although we age-standardised each country's sample to the world population, the lower age range in some surveys will underestimate risk, given increasing age is the strongest

predictor of elevated risk. Third, the majority of country surveys did not ascertain if participants had a prior CVD event, and given that the risk of a subsequent CVD event is considerably higher in this group, this will have underestimated the true risk profile. Fourth, although the proportion of missing data overall was small (4.1%) and the country-level median (IQR) proportion with missing outcome data across the 45 countries was 3.6% (1.8%–6.1%), there is potential for bias due to missing data, with this potential being greater in countries with higher proportions of missing data. Fifth, use of the non-laboratory-based equations may mean that some people, particularly those with diabetes, who were assessed as being at low risk would be reclassified to be at higher risk if laboratory values were available. This could lead to an overestimate in the proportion found to be taking BP medication without a clear indication, and, conversely, for those not on BP medication, it could lead to an underestimate of the treatment gap. Despite this important limitation, in many countries laboratory cholesterol values are not routinely ordered, and assessment of risk is dependent on non-laboratory-based equations. Also, given that STEPS and other surveys generally do not routinely collect laboratory measures, non-laboratory-based estimates are generally the only means by which population-representative risk estimates can be obtained.

If current trends continue, most LMICs are unlikely to achieve Sustainable Development Goal 3.4, to reduce premature mortality from NCDs by one-third by 2030 [27]. The case for investing in CVD prevention is therefore greater than ever. The WHO estimated that a US $120 billion investment in 20 high-CVD-burden countries (amounting to $1.50 per capita over 15 years) would avert 15 million deaths and 21 million incident ischaemic heart disease or stroke incidents in these countries [4]. This investment includes funding population-level strategies to lower CVD risk and strategies to improve healthcare performance. Although undoubtedly increased investment is needed, this study has shown that considerable improvements in health system performance could be achieved by better harnessing existing resources to appropriately use guidelines for managing CVD. Systematic CVD risk screening programmes from around 35–45 years of age have been shown to be more cost-effective than universal screening [7,28]. Although most countries report the availability of the basic requirements for non-laboratory-based CVD risk assessments in primary healthcare facilities, there remain major barriers to implementing a risk-based approach [29]. Implementation programmes to improve identification of people at high CVD risk in the community, intensify efforts to target subgroups that may be at elevated CVD risk, and shift treatments away from people at lower levels of risk towards these higher CVD risk groups may have the greatest potential to generate benefit both at the individual and health system levels [30].

The large heterogeneity of the findings in this study reflects varying country contexts. Factors such as subnational wealth distribution, overall health system expenditure, health system capability, and the shifting dynamics of CVD risk population profiles over time vary substantially across countries. Hence a one-size-fits-all approach is most likely not appropriate, and it would be necessary for international guidelines such as WHO PEN to incorporate country-specific modifications to ensure context specificity [7].

Although most country and international guidelines recommend CVD risk estimation as a critical first step in determining treatment decisions, this approach appears to be loosely adhered to in most countries in this study. Improved adoption of risk-based guidelines requires a substantial change in management approach at multiple levels in the health system. This includes targeted policies that are responsive to each country's context, engagement with professional bodies for workforce training and education, improved data systems and decision support that can be readily used by varying cadres of health workers, quality improvement processes in health facilities to enhance care effectiveness, and strategies to increase access and adherence to recommended medicines.

## Supporting information

**S1 STROBE Checklist.**
(PDF)

**S1 Fig. Percent with missing data for CVD risk estimation by country.**
(DOCX)

**S2 Fig. CVD risk profile for people aged 40–64 years.**
(DOCX)

**S3 Fig. Relative change (%) in 10-year CVD risk for females by educational level, household wealth, marital status, and employment status.**
(DOCX)

**S4 Fig. Relative change (%) in 10-year CVD risk for males by educational level, household wealth, marital status, and employment status.**
(DOCX)

**S5 Fig. Risk ratios (%) for taking BP medication by educational level, household wealth, marital status, and employment status, for individuals indicated for medication per World Health Organization/International Society of Hypertension guidelines.**
(DOCX)

**S6 Fig. Risk ratios (%) for taking BP medication by educational level, household wealth, marital status, and employment status, for individuals not indicated for medication per World Health Organization/International Society of Hypertension guidelines.**
(DOCX)

**S1 Table. Summary of surveys included.**
(DOCX)

**S2 Table. Sample characteristics among those excluded due to a missing outcome variable.**
(DOCX)

**S3 Table. Risk factor distribution by educational level, household wealth, marital status, and employment status, among males and females across the 4 regions.**
(DOCX)

**S1 Text. Computation of household wealth quintile.**
(DOCX)

**S2 Text. Statistical analysis plan.**
(DOCX)

**S3 Text. Data access information.**
(DOCX)

## Acknowledgments

We would like to thank each of the country-level survey teams and study participants who made this analysis possible.

## Author Contributions

**Conceptualization:** David Peiris, Arpita Ghosh, Lindsay M. Jaacks, Rifat Atun, Sebastian Vollmer, Justine I. Davies, Pascal Geldsetzer.

**Data curation:** Arpita Ghosh, Lindsay M. Jaacks, Michaela Theilmann, Maja E. Marcus, Zhaxybay Zhumadilov, Lindiwe Tsabedze, Adil Supiyev, Bahendeka K. Silver, Abla M. Sibai, Bolormaa Norov, Mary T. Mayige, Joao S. Martins, Nuno Lunet, Demetre Labadarios, Jutta M. A. Jorgensen, Corine Houehanou, David Guwatudde, Mongal S. Gurung, Albertino Damasceno, Krishna K. Aryal, Glennis Andall-Brereton, Kokou Agoudavi, Briar McKenzie, Jacqui Webster.

**Formal analysis:** David Peiris, Arpita Ghosh, Jennifer Manne-Goehler, Justine I. Davies, Pascal Geldsetzer.

**Methodology:** David Peiris, Arpita Ghosh, Jennifer Manne-Goehler, Lindsay M. Jaacks, Michaela Theilmann, Maja E. Marcus, Rifat Atun, Till Bärnighausen, Sebastian Vollmer, Justine I. Davies, Pascal Geldsetzer.

**Project administration:** Briar McKenzie, Jacqui Webster.

**Supervision:** David Peiris.

**Validation:** Michaela Theilmann.

**Writing – original draft:** David Peiris.

**Writing – review & editing:** Arpita Ghosh, Jennifer Manne-Goehler, Lindsay M. Jaacks, Michaela Theilmann, Maja E. Marcus, Zhaxybay Zhumadilov, Lindiwe Tsabedze, Adil Supiyev, Bahendeka K. Silver, Abla M. Sibai, Bolormaa Norov, Mary T. Mayige, Joao S. Martins, Nuno Lunet, Demetre Labadarios, Jutta M. A. Jorgensen, Corine Houehanou, David Guwatudde, Mongal S. Gurung, Albertino Damasceno, Krishna K. Aryal, Glennis Andall-Brereton, Kokou Agoudavi, Briar McKenzie, Jacqui Webster, Rifat Atun, Till Bärnighausen, Sebastian Vollmer, Justine I. Davies, Pascal Geldsetzer.

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
