## [Editor Report · Decision Letter 0]

29 Jun 2020

Dear Dr Peiris, 

Thank you for submitting your manuscript entitled "Cardiovascular disease risk profile and management practices in 45 low-income and middle-income countries: a cross sectional study of nationally representative individual-level survey data from 600,484 adults" for consideration by PLOS Medicine.

Your manuscript has now been evaluated by the PLOS Medicine editorial staff [as well as by an academic editor with relevant expertise] and I am writing to let you know that we would like to send your submission out for external peer review.

Kind regards,

Adya Misra, PhD,

Senior Editor

PLOS Medicine

---

## [Decision Letter · Decision Letter 1]

8 Sep 2020

Dear Dr. Peiris,

Thank you very much for submitting your manuscript "Cardiovascular disease risk profile and management practices in 45 low-income and middle-income countries: a cross sectional study of nationally representative individual-level survey data from 600,484 adults" (PMEDICINE-D-20-02958R1) for consideration at PLOS Medicine. 

[LINK]

In light of these reviews, I am afraid that we will not be able to accept the manuscript for publication in the journal in its current form, but we would like to consider a revised version that addresses the reviewers' and editors' comments. Obviously we cannot make any decision about publication until we have seen the revised manuscript and your response, and we plan to seek re-review by one or more of the reviewers. 

We expect to receive your revised manuscript by Sep 29 2020 11:59PM. Please email us (plosmedicine@plos.org) if you have any questions or concerns.

We look forward to receiving your revised manuscript. 

Sincerely,

Adya Misra, PhD

Senior Editor 

PLOS Medicine

plosmedicine.org

The background section of the abstract should clearly state the aim of the study

Please Provide participant demographics in the abstract

The last sentence of the methods and findings section should outline 2-3 limitations of your study design/methodology

Conclusions should be tempered with “our study shows….” or similar 

Data sharing: b) If the data are owned by a third party but freely available upon request, please note this and state the owner of the data set and contact information for data requests (web or email address). Note that a study author cannot be the contact person for the data.

Author summary

Perhaps introduce CVD and LMIC on first view for the author summary?

Introduction

References- please place these in square brackets, placing the punctuation after the bracket

The section “role of the funding source” should be removed from the main text and placed in the financial information section of the article meta-data

Please present and organize the Discussion as follows: a short, clear summary of the article's findings; what the study adds to existing research and where and why the results may differ from previous research; strengths and limitations of the study; implications and next steps for research, clinical practice, and/or public policy; one-paragraph conclusion.

Please add the following statement, or similar, to the Methods: "This study is reported as per the Strengthening the Reporting of Observational Studies in Epidemiology (STROBE) guideline (S1 Checklist)."

Did your study have a prospective protocol or analysis plan? Please state this (either way) early in the Methods section.

Comments from the reviewers:

Reviewer #1: In this study, Peiris and colleagues use cross-sectional data from ~600K adults across 45 countries to evaluate CVD 10-year risk and correlates with sociodemographic factors and blood pressure control.

My major question is the degree to which this study provides new information beyond the information provided in 2019 by the WHO CVD Risk Chart Working Group (S. Kaptoge et al, 2019), as Kaptoge et al report 10-year CVD risk across 21 global regions. Indeed, the work by Kaptoge et al highlight what I believe is a main limitation of this manuscript, namely the inferior performance of the non-laboratory risk equation. Indeed, Kaptoge describe the pragmatic models used herein as useful pre-selection tools given their poor performance among participants with diabetes. What are the implications for this finding for this manuscript, considering high global prevalence of diabetes (~9%) that shows marked heterogeneity by country? The decision to include participants <40 years of age also was poorly motivated and potentially erroneous, particularly since the equations are not calibrated for this age-group. It is therefore difficult to know how inclusion of these individuals, who are likely at very low CVD risk, affected the results, although one can speculate that they shifted countrywide estimates of 10-year CVD risk downwards. I also agree with the authors when then speculate that the inability to exclude prevalent cases likely biased their results downward. Yet, the prevalence of CVD likely varies across countries, complicating country-specific comparisons. The degree to which inclusion of prevalent cases affects cross-sectional associations reported in Figure 2 also is difficult to ascertain. 

Together, the limitations described above affect efforts to ascertain medicine overuse and underuse vis-à-vis anti-hypertensive agents. Limitations of the risk score aside, how much of the apparent discrepancy between medication usage and risk factor profile is captured by measurement error in blood pressure? These limitations call into question the authors statement that "the majority of BP medication is being used by people at lower levels of CVD risk."

Minor comments include conflation of CVD incidence and CVD mortality in the introduction as well as statements suggesting that leading risk factors for CVD can be diagnosed and treated at relatively low cost. (This statement ignores the rising tides of diabetes and obesity.) Finally, this paper seems to take a "prevention by treatment of high-risk individuals approach" as opposed to more population-wide approaches advocated by Geoffrey Rose, among others. The degree to which targeting high-risk individuals in resource strained settings remains unclear. 

Reviewer #2: I confine my remarks to statistical aspects of this paper. The general method is fine, but I have some issues and questions to resolve before I can recommend publication.

p. 7 Line 3 "Leading cause" is almost meaningless as it depends on how causes are defined. Give rates of death and disability. 

p. 11 line 8 Do you mean "simple linear regression"? (That is, one IV and one DV)? It's also not clear what you mean by "primary" --- do you mean "initial"? "Preliminary?" If you mean a process of bivariate screening, that is not a good method of building a model. 

 line 11 "multivariate" should be "multiple"

 line 12, 13 What covariance pattern was used?

 Line 15 Don't use quintiles and, if you have to use quintiles, don't combine them. The best would be to use an index. Categorizing continuous IVs is almost always a mistake. In *Regression Modelling Strategies* Frank Harrell lists 11 problems with this and sums up "nothing could be more disastrous". 

 line 17 Why was Poisson regression used? For dichotomous outcome variables, the usual choice is logistic regression

p. 12 line 20 "increased" is too causal a term. You can use "was higher" or something similar

Figure 1 - stacked bar plots are not a good method. See the work of William S., Cleveland. Here, I think density plots would be better (and that avoids categorizing the risk)

Figure 2 - The axes for the different variables should all b the same scaleF and wealth should not be categorized.

Figure 3 is a "dynamite plot" Theese are not recommended. A Cleveland dot plot may be better

Peter Flom

Reviewer #3: This is a well written paper which attempts to answer a set of important research questions with implications for NCD management improvement and Global NCD mortality reduction. Specifically, the authors use WHO STEP and related data from 600,484 individuals in 45 countries to:

1. explore the cardiovascular disease risk profile across and within these countries

2. assess the use of blood pressure medications as a surrogate for adherence to WHO PEN guidelines and to explore NCD management patterns 

3. assess the relationship of this CVD risk and management patterns to wealth, educational status and other sociodemographic factors

Major Comments

The rationale the authors present for the need for the study and the presence of a significant knowledge and/or gap that the research questions can address is not clear and compelling.

1. The authors rationale for the study is based on a number of assumptions that have implications on the inevitable findings, but for which little evidence is provided to support. 

a. The first is that WHO PEN is the most widely used technical guidance document in LMICS and therefore the guidance that most influences primary care practice. Unfortunately, we are not provided with any evidence that this is true. The evidence that is provided by the authors (references) is that the PEN guidelines are feasible but not very effective where used (in part because the uptake was poor). 

b. The second assumption is that most countries and health systems use the WHO PEN or GLOBAL HEARTS approach, which recommend CVD risk estimation-for decisions on treatment interventions for interventions. 

Many (perhaps most) countries across sub Saharan Africa, Indian sub-Continent Asia, Middle East and South America also have local or regional Blood Pressure Management guidelines and or access to WHO and International Society of Hypertension (or combined) guidelines on the management of blood pressure. 

In contrast to the WHO PEN guidance, almost all of these blood pressure guideline documents dating back to the early 2000s stipulate that if "low risk" hypertensive patients do not respond to lifestyle interventions (diet, weight loss, exercise) within 3-6 months, they should be treated with an antihypertensive agents. This is important because if it is true that most of the globe integrates separate hypertension guidelines to management in their primary health care approaches, the research question may be somewhat rhetorical i.e., at attempt to prove what we "know" rather than find out what we do not know. There is also a distinct possibility that the reason for one of their main findings -that low risk patients are "overtreated" may be by design-i.e., because the health system and health care practitioners are adhering in part to local guidance on BP treatment and not lack of adherence to WHO guidance. The 47% of people on BP treatment who they find do not have an indication for BP may infect have a clear indication based on current and former hypertension guidelines. 

The methods and analytical approach to address the questions are appropriate. 

The results and their interpretation are sound. The only question is that one may ask is whether the study has addressed any significant gaps in knowledge or strengthened the available evidence

The discussion section and conclusion are solid and are a fair reflection or interpretation of the results 

In summary I would recommend that the issues raised around the rationale and assumptions need to be addressed prior to accepting the paper

[LINK]

---

## [Decision Letter · Decision Letter 2]

9 Nov 2020

Dear Dr. Peiris,

Thank you very much for re-submitting your manuscript "Cardiovascular disease risk profile and management practices in 45 low-income and middle-income countries: a cross sectional study of nationally representative individual-level survey data from 600,484 adults" (PMEDICINE-D-20-02958R2) for review by PLOS Medicine.

I have discussed the paper with my colleagues and the academic editor and it was also seen again by xxx reviewers. I am pleased to say that provided the remaining editorial and production issues are dealt with we are planning to accept the paper for publication in the journal.

[LINK]

We look forward to receiving the revised manuscript by Nov 16 2020 11:59PM. 

Sincerely,

Adya Misra, PhD

Senior Editor 

PLOS Medicine

plosmedicine.org

Requests from Editors:

I suggest the title is shortened to: “Cardiovascular disease risk profile and management practices in 45 low and middle-income countries: a cross sectional study of individual-level survey data”

Data availability- please provide a link to the data analysis code

Abstract Line 9 should contain “help”

Please avoid "almost half" in the abstract - the actual proportion given will suffice.

In both abstract and author summary, we suggest removing the word "relative" from "relative over-use".

Page 12- please remove the section “role of the funding source” from the main text. This information is pulled from the article meta-data. Please also change this in the STROBE checklist.

Please provide your pre-specified analysis plan as SI files and provide a call out to this in the methods section

Page 16 line 15 I suggest “hypertensive patient” is changed to “patient with hypertension”

Please remove the underlining early in the Introduction.

Please ensure that all reference call-outs fall before punctuation, and remove spaces from the square brackets (e.g., "... [12,13].").

Please add an individual or institutional author name to reference 1

References need to be formatted to Vancouver style. Please check ref 15 and revise as needed

Author summary point 1, I'd the burden is high, not "large"

Author summary, last point, could remove "A one size fits all approach is unlikely to be appropriate and"

Ethics statement- please add a brief sentence to reassure readers that the original researchers who put together the datasets used here obtained ethics approval and informed consent.

Please restate SDG 3.4 when they mention it in the Discussion on page 17

Comments from Reviewers:

Reviewer #1: The authors state that they included additional data when compared to Kaptoge et al., 2019. What proportion of their data overlapped and how specifically are these data different? Regarding blood pressure measurement error, I believe the authors misunderstood the spirit of my question, as blood pressure reliability is moderate at best. Thus, the authors' assertion that BP meds are being used by people at lower CVD risk levels might simply reflect the inability of one measure of blood pressure to capture elevated blood pressure assessed by care providers over a much longer time. 

Reviewer #2: The authors have addressed my concerns and I now recommend publication

Peter Flom

Reviewer #3: I am satisfied that the comments and queries raised in the initial review have been adequately addressed

[LINK]

---

## [Editor Report · Decision Letter 3]

1 Feb 2021

Dear Dr. Peiris,

I am writing concerning your manuscript submitted to PLOS Medicine, entitled “Cardiovascular disease risk profile and management practices in 45 low-income and middle-income countries: a cross sectional study of nationally representative individual-level survey data”.

We have now completed our final technical checks and have approved your submission for publication. You will shortly receive a letter of formal acceptance from the editor.

Kind regards,

PLOS Medicine